# ROBOTICS IN REPRESENTATION SPACE: LEARNED LATENTS MEET COMPOSABLE COSTS

## ABSTRACT

Deep learning methods have vastly expanded the capabilities of motion planning in robotics applications, as learning priors from large-scale data has shown to be essential in capturing the highly complex behavior required for solving tasks such as manipulation or navigation for autonomous vehicles. At the same time, model-based planning algorithms based on search or optimization remain an essential tool due to their flexibility, efficiency and the ability to incorporate domain knowledge via expert designed algorithms and objective functions. We propose a simple framework to unify these two paradigms. First, we learn an autoencoder with a high compression ratio and a latent space of causally ordered, discrete-valued tokens. Leveraging both the dimensionality reduction and the causal structure learned by this autoencoder, we then perform motion planning by directly searching in the latent space of tokens. Notably, this search can optimize arbitrary user-specified objective functions without requiring the training of any additional neural networks, providing a large degree of flexibility at test time while maintaining efficiency and producing feasible and realistic solutions by relying on the generative capabilities of the highly compressed autoencoder. We evaluate our method on the Waymo Open Motion Dataset, showing how a simple latent space search can be used for motion prediction. Beyond prediction, we demonstrate the inclusion of simple objectives for guided behavior generation. Finally, we investigate the application of our method for multi-agent interaction modeling, enabling flexible scenario design and understanding.

## 1 INTRODUCTION

In scaling image generation to ever increasing sample quality and resolution, *compression* has been a key enabler. Indeed, state-of-the art image generative models such as latent diffusion (Rombach et al., 2022) or autoregression (Chang et al., 2022; Li et al., 2024) typically operate in the space of tokens learned with an autoencoder (Kingma & Welling, 2014; van den Oord et al., 2017; Esser et al., 2021). Crucially, the autoencoder is able to exploit the high degree of redundancy present in natural images to produce a latent space of tokens which is much lower dimensional than the original pixel space (Rombach et al., 2022). Since a more highly compressed representation directly results in lower-dimensional data for the generative model to predict, there has therefore been interest in not only improving the generative model itself, but also the autoencoder used as the tokenizer.

For example, Yu et al. (2024) represent $256\,\mathrm{px} \times 256\,\mathrm{px}$ ImageNet images (Deng et al., 2009) with as few as 32 discrete-valued tokens, allowing an image generative model to predict new samples very efficiently by operating in this compact latent space. But what happens when the compression ratio provided by the autoencoder is scaled higher and higher? In the case of image generation, Lao Beyer et al. (2025) find that with increasing responsibility shifted to the token *decoder*, the generator's task becomes so easy that it can be replaced with simple heuristic token manipulation or prediction techniques which do not require training of a dedicated generative model at all.

Inspired by the success in scaling image tokenizers and the training-free generation capabilities enabled by highly compressed tokenized representations, in this paper we explore the value of autoencoders with very high compression ratios in robotics applications. In robotics, motion planning has traditionally relied heavily on a classical toolbox including, for example, trajectory optimization. These classical approaches can typically incorporate objectives and constraints designed by domain

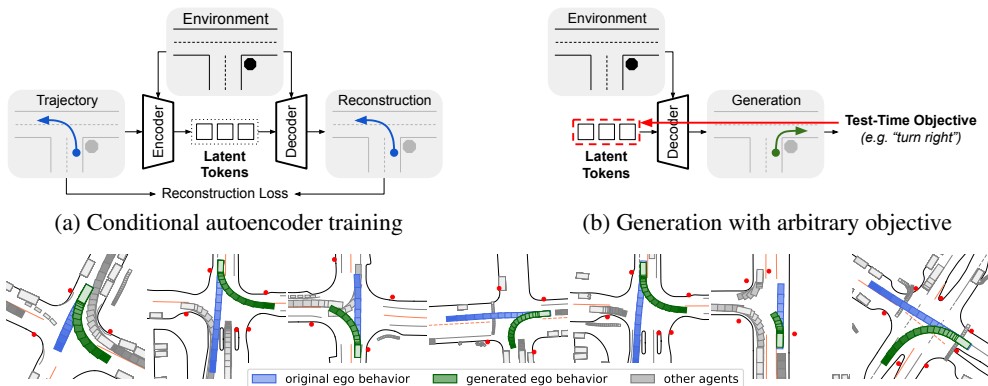

(a) Conditional autoencoder training  (b) Generation with arbitrary objective

(c) Token search synthesizes desired behavior (left turn; green) differing from the original (straight; blue).

Figure 1: **Rich and compact latent representation enables environment-conditioned generation with flexible test-time objectives.** A conditional autoencoder is trained with a reconstruction objective to capture a highly compact latent representation of the input trajectory given a particular environment *(a)*. Direct search in the latent space of tokens is used to generate desired behavior at test-time, without additional training — *(b)* and *(c)*.

experts, and can be highly robust and performant when deployed in controlled settings (Foehn et al., 2021; Moore et al., 2014; Goh & Gerdes, 2016). However, with autonomous systems operating in increasingly more unstructured and open-ended domains, it has become apparent that learning powerful priors from large-scale, real-world data is necessary. We therefore argue that *generation as direct search over latent tokens is especially useful in robotics tasks* as it provides a framework for combining deep priors (in the form of a powerful token decoder) with model-based objectives (optimized by performing search in the latent space of tokens).

In our paper, we show search in the latent token space of a trajectory autoencoder can indeed be used to optimize arbitrary user-defined objectives. To facilitate efficient implementation of this search, we present an environment-conditioned trajectory autoencoder that learns highly compressed, discrete, and causally-ordered variable length trajectory representations. We train our autoencoder the Waymo Open Motion Dataset (Ettinger et al., 2021) and demonstrate its prediction and planning capabilities when paired with an efficient greedy latent space search.

## 2 CONDITIONAL TRAJECTORY AUTOENCODER

Trajectory prediction and planning problems in robotics must consider information such as sensor inputs or known maps. We therefore choose to represent a trajectory *conditionally* on given information about the environment. To this end, we train a conditional autoencoder with the goal of learning a compact and expressive learn latent representation of the trajectory under a particular fixed environment (see Figure 1).

**Notation and dataset.** We denote the environment as $\mathcal{E}$. In the case of the Waymo Open Motion Dataset (WOMD) (Ettinger et al., 2021), $\mathcal{E}$ consists of static world features such as road edges, road lines, lane geometry, stop signs and traffic lights, as well as one second of dynamic object history in the form of 10 past and one current observation for each visible agent in the scene (including the ego agent). The full trajectory of the agent of interest is denoted as $\mathcal{T}$. In the setting of motion prediction in WOMD, this full trajectory is represented as 80 position samples corresponding to an 8 second long future trajectory. The encoder Enc then produces a sequence of $N$ $D$-dimensional latent tokens $z := \{\mathbf{z}_i \in \mathbb{R}^D\}_{i=1}^N = \text{Enc}(\mathcal{T}, \mathcal{E})$ from the full trajectory $\mathcal{T}$ of the agent of interest and the environment $\mathcal{E}$. The decoder Dec attempts to reconstruct the trajectory $\mathcal{T}$ from the latent tokens and the environment, producing a prediction $\mathcal{T}_{\text{pred}} = \text{Dec}(\mathbf{z}, \mathcal{E})$.

**Prediction space and loss function.** The decoder Dec predicts mean and variance parameters of a Gaussian distribution for each point along the trajectory, and is trained by minimizing the negative log-likelihood (NLL) of the ground truth reconstruction under the prediction. In practice, we make use of the $\beta$-NLL (Seitzer et al., 2022) due to its improved convergence characteristics compared to the standard NLL objective.

## 2.1 COMPRESSION VIA ADAPTIVE SOFT QUANTIZATION

Recent work on image generation shows that vector quantization is essential in allowing direct latent space search and manipulation to produce meaningful outputs (Lao Beyer et al., 2025). Indeed, when the latent space is too expressive, there exist many encodings which the decoder can map to outputs that do not lie on the desired data manifold. Direct search in such an overly expressive latent space instead behaves adversarially, unless a robust objective is used (Santurkar et al., 2019).

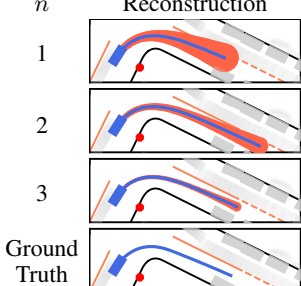

Figure 2: **Adaptive noise injection outperforms fixed noise level.** Note that validation ADE is lower than training ADE since during validation $\sigma_t = 0$.

To mitigate training challenges commonly associated with vector quantization while maintaining its desirable regularizing effects we use a form of "soft" quantization consisting of noise injection at the autoencoder bottleneck:

$$\texttt{corrupt}(\mathbf{z}) = \tanh(\mathbf{z}) + \boldsymbol{\epsilon}_t \quad \text{with} \quad \boldsymbol{\epsilon}_t \sim \mathcal{N}(0, I\sigma_t^2). \quad (1)$$

Note that a $\tanh$ activation is applied pointwise to the input, effectively creating an amplitude-limited noisy channel. The chosen noise level $\sigma_t^2$ is picked adaptively during training to gradually ramp up from zero until a desired reconstruction accuracy is achieved. Concretely, we consider the average displacement error (ADE) averaged across every prediction in the current training minibatch $\text{ADE}_t$ to adjust $\sigma_t$:

$$\sigma_t = \max(0, \gamma\sigma_{t-1} + (1-\gamma)\hat{\sigma}_t) \quad \text{with} \quad \hat{\sigma}_t = \begin{cases} \sigma_{t-1} + \Delta\sigma & \text{ADE}_t \leq \text{ADE}_{\text{target}} \\ \sigma_{t-1} - \Delta\sigma & \text{ADE}_t > \text{ADE}_{\text{target}} \end{cases}. \quad (2)$$

Here, $\text{ADE}_{\text{target}}$ denotes the target training ADE and is a fixed hyperparameter. Likewise, the decay factor $\gamma \in [0, 1)$ and the noise increment $\Delta\sigma > 0$ can be used to tune the responsiveness of the adaptive schedule to changes in $\text{ADE}_t$. During training, we apply $\texttt{corrupt}$ to each token $\mathbf{z}_i$ before feeding it to the decoder. At test time, we set $\sigma_t = 0$. As illustrated in Figure 2, we find that this adaptive noise schedule outperforms choosing a fixed noise level.

We refer to this process as soft *quantization* since our $\texttt{corrupt}$ procedure resembles an amplitude-limited Gaussian channel, for which the input distribution achieving maximum information capacity is known to be discrete (Smith, 1971).

**Hard quantization at test time.** At test time, we may leverage the decoder's ability to predict starting from heavily noised input tokens by explicitly quantizing the encoder's output. For this purpose, we round each element $(\mathbf{z}_i)_j$ of each token $\mathbf{z}_i$ to the nearest quantized level $L[k] \in \{-1, \dots, 1\}$ out of the $N_{\text{levels}}$ uniformly spaced options. Note that $\mathbf{z}_i$ here refers to the token after applying the $\tanh$ activation from Equation (1).

## 2.2 VARIABLE-LENGTH LATENT ENCODING

To allow flexible reconstruction fidelity at test time and to facilitate structured exploration of the latent space, we choose to impose a causal ordering structure on the latent tokens $\mathbf{z}_i$. Causality among the tokens $\mathbf{z}_i$ is enforced via causally masked self-attention in the encoder and decoder networks. We additionally make use of nested dropout (Rippel et al., 2014) to allow the decoder to operate on variable-length encodings. During training, nested dropout drops a random number $m < N$ of tail tokens from the encoding, forcing the decoder to reconstruct the trajectory from only the first $n := (N - m)$ tokens.

Figure 3: Using more tokens $n$ results in better reconstructions (blue) with lower predicted uncertainty (red).

As highlighted in Figure 3, at test time, latent token sequences of any size may be used to obtain reconstructions of varying degrees of fidelity, as causal masking combined with nested dropout ensures that later tokens capture increasingly more fine-grained information. For planning tasks, this property suggests a *greedy* latent space search strategy in which tokens can be picked one at a time. We explore this idea in Section 3.2.

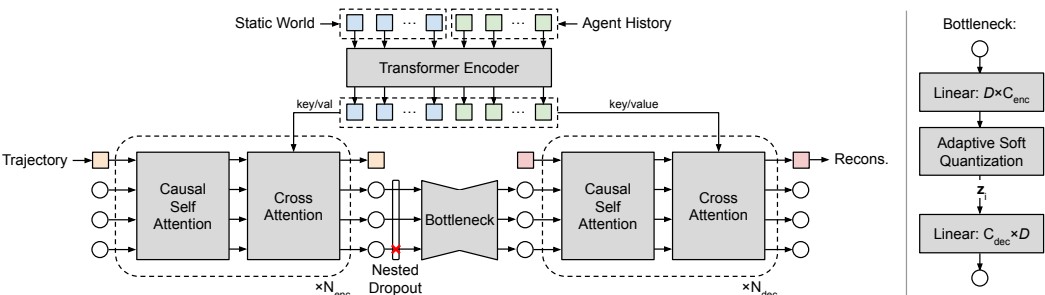

Figure 4: **Conditional autoencoder architecture.** The environment (static world, dynamic object histories) is tokenized and fed into a transformer encoder consisting of local neighborhood self-attention layers. The processed tokens form the key and value for the cross-attention-based encoder and decoder transformers. Encoder and decoder impose causal masking among latent tokens.

## 2.3 NETWORK ARCHITECTURE

Our conditional autoencoder is implemented using three transformer models: an environment encoder which processes static world and dynamic object history information using local neighborhood self-attention layers, and architecturally identical encoder and decoder models which attend to the environment information via cross-attention and process the latent tokens via causal self-attention. Our environment encoder follows Motion Transformer (MTR), making use of MTR's local neighborhood attention (Shi et al., 2022) in which self-attention between environment features is restricted to 16 nearest neighbors for efficiency. Similarly to MTR, static world and object history are tokenized from their vectorized representation as collections of polylines via PointNet encoders (Qi et al., 2017). We also apply a PointNet encoding to the input trajectory fed to the encoder. An MLP regresses the output trajectory $\mathcal{T}_{\text{pred}}$ from a [CLS] query token processed by the decoder. At the bottleneck between encoder and decoder transformer models, we first project the transformer tokens down to the desired low dimensionality $D$, during training apply nested dropout and noise corruption for soft quantization, and finally project the low-dimensional tokens back to the transformer token dimensionality. An overview of this design is shown in Figure 4.

## 3 BEYOND RECONSTRUCTION: TOWARD PREDICTION & PLANNING

In this section we demonstrate the generative capabilities of our conditional autoencoder. We train the autoencoder on the single-agent trajectory reconstruction task, choosing a highly compressed bottleneck with maximum number of tokens $N = 3$, token dimensionality $D = 3$, and continuing training until a high soft quantization noise of $\sigma_t > 0.35$ is achieved with $\text{ADE}_{\text{target}} = 0.65$.

### 3.1 TOKEN SEMANTICS ENABLE BEHAVIOR TRANSFER

We show that latent token encodings learned by our conditional autoencoder carry significant high-level semantic information through a series of simple experiments.

**Token swapping.** Consider decoding the latent token representation of a given trajectory $\mathcal{T}_A$ in its environment $\mathcal{E}_A$ under a *different* environment $\mathcal{E}_B$:

$$\mathcal{T}_{A \to B} = \texttt{Dec}(z_A, \mathcal{E}_B) \quad \text{with} \quad z_A = \texttt{Enc}(\mathcal{T}_A, \mathcal{E}_A). \tag{3}$$

Figure 5a shows the result of this simple manipulation for selected scenarios where the agent of interest is a vehicle at an intersection. We observe that decoding an encoding corresponding to a desired behavior in a different scenario can be used to *transfer* this behavior to a novel scenario.

**Behavior transfer across scenarios.** We now wish to verify whether token sequences corresponding to certain high-level behavior can be automatically identified and transferred to a large number of alternate environments. For this purpose, we collect a subset of of the WOMD test set's scenarios satisfying some condition (for example, proximity to several stop signs and travel at low speeds, often corresponding to scenarios with all-way stop intersections). We then further divide this subset into buckets corresponding to classes of maneuvers (for example, turning direction). For each such

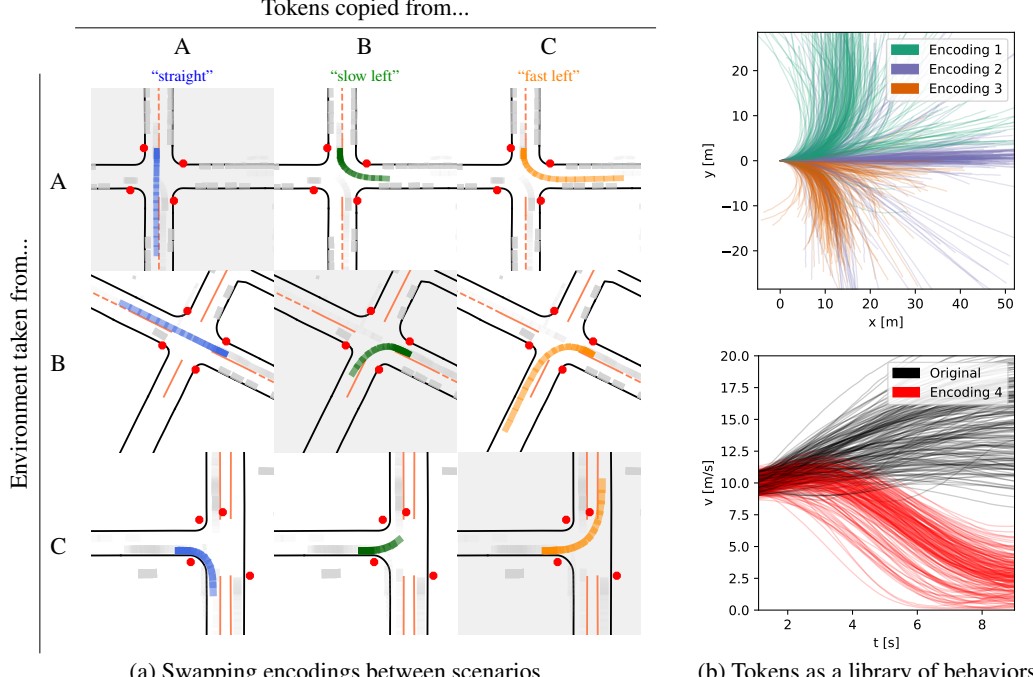

(a) Swapping encodings between scenarios          (b) Tokens as a library of behaviors

Figure 5: **Tokens have meaningful environment-dependent semantics.** When copying the encoding of a given trajectory under a particular environment of the WOMD test set and decoding it under a *different* test environment (Equation (3)), predictable behavior consistent with the new environment is produced. Shaded plots in *(a)* show the reference trajectory reconstructed in its original environment, while the remaining plots correspond to decoding the tokens of a particular behavior in an environment other than the one used to encode them. Note that Environment C does not admit driving straight, and in this case the decoder produces a valid alternative reconstruction (last row, first column). In *(b)*, we decode each latent token encoding from a small pre-selected library (Encoding 1, 2 or 3) in ~250 WOMD test set environments containing intersections (top). We also select an additional Encoding 4 corresponding to a high deceleration event, and plot the speed profiles resulting from decoding this encoding in ~200 environments of similar starting speed alongside the original speed profiles (bottom).

bucket, we compute the most common (discrete) token encoding, yielding a single token sequence for each bucket. Finally, we decode each encoding in every environment. The results presented in Figure 5b strongly suggest that a class of maneuvers may be characterized by a single latent token sequence, and the corresponding behavior may be transferred to new environments by simply decoding these tokens conditioned on that new environment.

## 3.2 LATENT TOKEN SEARCH

The combination of low-dimensional ($D = 3$), highly quantized ($\sigma_t > 0.3$) and causally ordered tokens suggests a very straightforward and efficient way to explore the latent space of our decoder: greedy best-first search.

**Hard quantization at test-time.** Before attempting discrete tree search over quantized tokens, we verify reconstruction accuracy under hard quantization. As shown in Table 1, we find that the conditional autoencoder's drop in reconstruction accuracy is relatively modest even when heavily quantizing the tokens to $N_{\text{levels}} = 2$. Note that increasing the number of tokens leads to better reconstructions compared to keeping the number of tokens fixed and using finer quantization, even when the theoretical capacity of a larger number of more coarsely discretized tokens is lower.

**Greedy tree search.** To check whether greedy best-first search is a suitable strategy for exploring the space of latent tokens, we consider a ground-truth reconstruction objective. This simple algorithm picks tokens one at a time, by evaluating an ADE reconstruction objective wrt. the ground truth trajectory. In particular, it evaluates the decoder output corresponding to every possible quan-

Table 1: **Reconstruction with greedy search outperforms the learned encoder.** Greedy search can match or exceed the reconstruction performance of the learned encoder even when not applying any hard quantization in the autoencoder bottleneck. We use the same ADE metric averaged over object types and prediction horizons as the WOMD prediction challenge.

| | Average Absolute Deviation Error (ADE) $\downarrow$ | | | | |
| Num. Tokens | Autoencoder | | | Greedy Search | |
| | $N_{\text{levels}} = 2$ | $N_{\text{levels}} = 3$ | no quant. | $N_{\text{levels}} = 2$ | $N_{\text{levels}} = 3$ |
|---|---|---|---|---|---|
| 1 | 0.800 | 0.617 | *0.567* | 0.708 | **0.524** |
| 2 | 0.519 | 0.410 | *0.365* | 0.485 | **0.363** |
| 3 | 0.403 | 0.334 | **0.298** | 0.386 | *0.301* |

Table 2: **Motion prediction via latent space search.** Even though our conditional autoencoder is trained to perform reconstruction instead of prediction, we find that search over latent tokens leads to high quality decoded trajectories. While not competitive with highly tuned state-of-the-art trajectory prediction methods, performance exceeds or approaches that of many common prediction baselines. Note that predicted variance is helpful in informing token selection, as the *random* objective function which assigns arbitrary confidence scores to each token results in degraded performance (last row). †: Results from the WOMD validation set due to submission number limitations for the test set.

| Model | $\text{minADE}_6 \downarrow$ | $\text{minFDE}_6 \downarrow$ |
|---|---|---|
| Waymo LSTM Baseline (Ettinger et al., 2021) | 1.0065 | 2.3553 |
| MotionCNN (Konev et al., 2022) | 0.7400 | 1.4936 |
| Scene Transformer Ngiam et al. (2022) | 0.6117 | 1.2116 |
| MTR (Shi et al., 2022) | 0.6050 | 1.2207 |
| DriveGPT (Huang et al., 2025) | 0.5240 | 1.0538 |
| **Decoder with variance minimization objective** | 0.6793 | 1.4291 |
| Decoder with variance minimization objective † | 0.6416 | 1.3882 |
| Decoder with random objective † | 0.7311 | 1.5954 |

tized value of the (single) next token, picking the best reconstruction at each iteration. The greedy search with reconstruction objective forms a valid replacement for the learned encoder in our autoencoder. Indeed, Table 1 shows that greedy search significantly outperforms the learned encoder, demonstrating that greedy token selection is a valid approach thanks to the causal and noise-resilient structure of the autoencoder's latent space.

## 3.3 PREDICTION

The token search presented in the previous section is not limited to reconstruction tasks. For example, we may want to consider the trajectory prediction task, in which we do not have access to the input trajectory $\mathcal{T}$. Recall that our model predicts not only the mean but also a variance associated with each sample in the predicted trajectory. Therefore, we consider searching for the tokens that will minimize the variance of the final sample in each trajectory.

**Results.** Despite being trained as an autoencoder with a reconstruction loss, we find that our model is able to achieve high quality prediction results when paired with the variance-minimizing latent space search. Table 2 summarizes our results against common baselines and demonstrates the effectiveness of variance minimization as an objective function by comparing with arbitrary token selection. Note that we have trained a model with a more highly compressed latent representation for this experiment by setting $N = 1$ and $D = 3$, and that we use $N_{\text{levels}} = 2$ for the test-time discretization.

## 3.4 PLANNING WITH ARBITRARY OBJECTIVES

While our motion prediction results are encouraging, the main utility of our framework lies not in its ability to perform prediction, but in the flexibility it affords in order to explore the space of possible behaviors efficiently and according to arbitrary objective functions specified at test time without requiring re-training of any models. To this end, we evaluate the effectiveness of latent space search for guided maneuver generation.

Table 3: **Maneuver optimization according to user specified objectives.** Greedy token search efficiently explores the conditional autoencoder's latent space with increasing success rate for increased search depth. Note that success rate is not expected to reach 100%, as datasets include cases where desired maneuver is impossible or illegal, such as turning left from a lane other than the dedicated left turn lane or requiring excessive acceleration or deceleration. *Edge contact* refers to the agent contacting static road edge geometry. Main text describes details on objectives and success metrics.

| | Left Turn Objective | | Speed Reduction Objective | |
| --- | --- | --- | --- | --- |
| Method | Success Rate | Edge Contact | Success Rate | Edge Contact |
| None (original scenario) | 0% | 0% | 0% | 0.76% |
| Token search (1 token) | 59.0% | 0% | 28.7% | 0.63% |
| Token search (2 tokens) | 72.6% | 0% | 55.4% | 0.38% |
| **Token search (all 3 tokens)** | **75.5%** | **0%** | **63.2%** | **0.13%** |

**Turn maneuver at intersection.** We automatically select $\sim$300 test set scenarios in which the agent of interest is a vehicle traveling straight and at low speed in proximity of at least four stop signs. This roughly corresponds to cases in which the vehicle is traversing an intersection and going straight. We consider an objective function which maximizes the cumulative leftward heading change along the trajectory while heavily penalizing excessive predicted variance. Success is defined as achieving a cumulative leftward heading change of over $45°$.

**Speed profile optimization.** We now consider $\sim$800 automatically selected test scenarios in which a vehicle is traveling at an initial speed of around $9\,\mathrm{m/s}$ and maintains a similar average speed throughout its full trajectory. In this case, our objective is to slow down to a lower final speed of $5\,\mathrm{m/s}$ maintained for the last three seconds of the trajectory, again while imposing a heavy penalty on predictions that are assigned high variance by the decoder.

Results for these experiments are presented in Table 3. With a high rate of success, token search is able to generate maneuvers according to the specification, while our token decoder automatically ensures that behavior is consistent with the given scenario as evidenced by zero or near-zero rates of contact between the predicted trajectory and road edge geometry. Further details and results on planning via latent space search may be found in the appendix (Section A.2).

**Performance.** With the parameters of $N = 3$, $D = 3$ and $N_{\text{levels}} = 2$ used for experiments in this section, greedy search requires just 24 evaluations of the decoder, which is exponentially less than the 512 evaluations that would be required to perform an exhaustive search. On the NVIDIA RTX 6000 Ada GPU, greedy search with these parameters can generate about 115 trajectories per second, corresponding to about 2760 decoder calls per second. Note that each call to the environment encoder is amortized across 24 decoder calls during search.

### 3.5 JOINT TRAJECTORY TOKENIZATION FOR MULTI-AGENT TASKS

So far, we have trained a conditional autoencoder to reconstruct a single agent's trajectory, and have then leveraged latent token manipulation or search for prediction and planning tasks for that single agent. However, joint tokenization of several agents' trajectories offers potential for learning even more informative and compact representations that exploit the correlations between trajectories present in multi-agent interactions. In this section we therefore investigate the application of our proposed framework to the multi-agent setting.

**Multi-agent modeling.** To construct the multi-agent conditional autoencoder, we leverage the single-agent encoder and decoder, which predict and decode, respectively, a single high-dimensional feature vector $\mathbf{y_a}$ corresponding to the trajectory of agent $a$. We then further encode the set of these feature vectors jointly using a second-stage encoder, in order to produce the final token sequence $z$. The corresponding decoder transforms the latent tokens $z$ back to the set of feature vectors $\{\mathbf{y_1}, \ldots, \mathbf{y_A}\}$, which are finally individually decoded by the single-agent decoder network. We implement this additional encoder and decoder using self-attention across the agent tokens $\mathbf{y_a}$ and the latent tokens. As in the single-agent case, we apply adaptive soft quantization via noise injection at training at the autoencoder bottleneck.

**Reconstruction.** We train our conditional autoencoder on scenarios with up to 8 agents using $N = 4$, $D = 3$ and $\text{ADE}_{\text{target}} = 0.8$, continuing training up to a noise level of $\sigma_t > 0.08$.

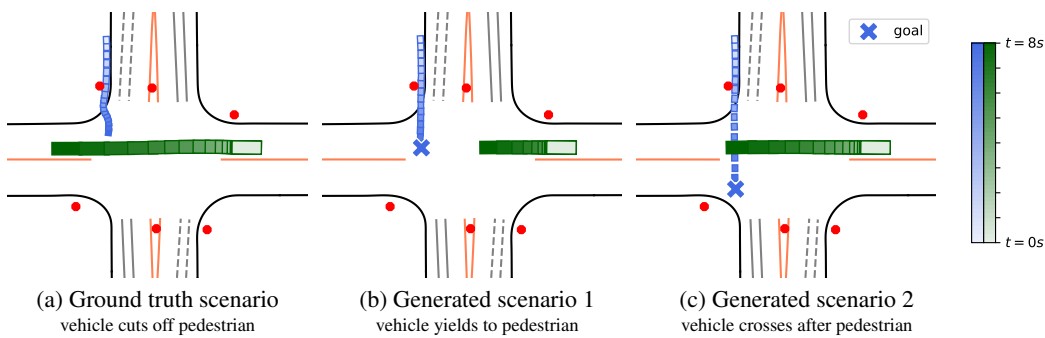

| (a) Ground truth scenario | (b) Generated scenario 1 | (c) Generated scenario 2 |
|:-:|:-:|:-:|
| vehicle cuts off pedestrian | vehicle yields to pedestrian | vehicle crosses after pedestrian |

Figure 6: **Multi-agent token search generates consistent joint trajectories.** We generate two alternate scenarios *(b)* and *(c)* for the environment from *(a)* by performing token search to minimize the deviation between the final position of the pedestrian (blue) and a user-specified goal point (cross marker). Even though this objective function only supervises the final position of the pedestrian, our joint trajectory decoder ensures that the behavior of the vehicle (green) is valid.

Table 4: **Multi-agent tokens enable semantic understanding.** With the latent token sequence learned by our conditional autoencoder as extra input, a language model (Qwen3-4B-Instruct-2507) fine-tuned on the WOMD-Reasoning dataset beats VLM baselines and roughly matches Motion-LLaVA on language metrics evaluating question answering performance.

| Model | ROUGE-L ↑ | BLEU ↑ | METEOR ↑ | CIDEr ↑ | SPICE ↑ |
|---|---|---|---|---|---|
| Non-fine-tuned LLaVA (Li et al., 2025) | 0.512 | 0.211 | 0.275 | 1.36 | 0.455 |
| Fine-tuned LLaVA (Li et al., 2025) | 0.779 | 0.581 | 0.439 | 5.51 | 0.735 |
| Motion-LLaVA (Li et al., 2025) | 0.792 | 0.616 | 0.449 | 5.69 | 0.744 |
| **Ours** | 0.788 | 0.611 | 0.450 | 5.68 | 0.724 |

As shown in Table 5, this enables reconstructing multi-agent scenarios from just four tokens quantized with $N_{\text{levels}} = 3$. We again verify that greedy token search can match the learned encoder for a given degree of quantization.

**Interaction Generation.** Leveraging the flexibility that latent token search provides in enabling arbitrary objective functions, in Figure 6 we impose a terminal goal position *for a single agent in the scene* and observe how the joint trajectory decoder ensures that *other* agents' behavior is adjusted.

Table 5: Multi-agent reconstruction.

| Num Tok | Average ADE ↓ | | |
|---|---|---|---|
| | Autoencoder | | Greedy |
| | $N_{\text{lvl}} = 3$ | no quant. | $N_{\text{lvl}} = 3$ |
| 1 | 1.093 | *1.055* | **1.029** |
| 2 | 1.001 | **0.882** | *0.968* |
| 3 | 0.950 | **0.756** | *0.934* |
| 4 | *0.880* | **0.663** | 0.886 |

**Interaction Understanding.** Just as in the single agent case (see Section 3.1), we find that latent tokens of our multi-agent conditional autoencoder carry high-level semantic information. We verify this by learning a small two-layer adapter MLP between the latent tokens produced by our encoder, and the embedding space of a pretrained large language model (LLM). The output of our environment encoder is also projected to the LLM's embedding space. We train the projection layers as well as a low-rank adaptation to the LLM weights (LoRA) (Hu et al., 2022) on the *WOMD-Reasoning* dataset (Li et al., 2025), which contains question and answer pairs related to WOMD scenarios. Note that we do not fine-tune the conditional autoencoder's tokens. Results of this experiment, which are presented in Table 4, indicate that an LLM — specifically, Qwen3-4B-Instruct-2507 (Qwen Team, 2025) — can match the performance of the *Motion-LLaVA* model (Li et al., 2025) on language metrics, when provided with our tokenized representation. In contrast to our fixed encoder, Motion-LLaVA is a dedicated multimodal motion understanding model based on LLaVA-v1.5-7b Liu et al. (2023) which is fine-tuned end-to-end, including the motion vector encoder.

## 4    RELATED WORK

**Autoencoders for image tokenization.** In image generation, state-of-the art generative models rely on tokenization, rather than operating directly in the pixel domain for efficiency reasons (Esser et al., 2021; Rombach et al., 2022; Chang et al., 2022; Yu et al., 2022b; Li et al., 2024). Com-

monly used tokenizers include variational autoencoders (VAEs) (Kingma & Welling, 2014) which learn continuous-valued tokens that can be modeled with diffusion. In order to facilitate autoregressive image modeling (Chang et al., 2022), another commonly used class of tokenizers includes the vector-quantized VAE (van den Oord et al., 2017) and VQGAN (Esser et al., 2021) which improves perceptual quality of reconstructions using an adversarial loss (Goodfellow et al., 2014). Vector quantized representations often pose challenges during training, such as codebook collapse, requiring the use of carefully tuned auxiliary losses (Yu et al., 2022a), and motivating the adaptive noise injection used in our model. We note also that tokenizers for image generation do not usually include any mechanism comparable to our environment conditioning, as conditional generation is typically handled by the generative model and not the tokenizer. One exception is TA-TiTok (Kim et al., 2025), which supports text-conditioned decoding of latent image representations.

**Variable-length tokenization and ordered representations.** Rippel et al. (2014) introduces nested dropout for learning ordered representations. More recently, several image tokenization models (Wen et al., 2025; Miwa et al., 2025; Bachmann et al., 2025) apply nested dropout for causally ordered variable-length image representations. These tokenized representations are found to effectively learn a "coarse-to-fine" latent image representation, which inspires our greedy latent exploration approach.

**Training-free generation via latent space search.** Gradient-based optimization has been used to optimize GAN latents according to a CLIP (Radford et al., 2021) objective for text-to-image generation (Patashnik et al., 2021). Operating in the space of an image tokenizer instead of a GAN, VQGAN-CLIP (Crowson et al., 2022) optimizes tokens according to a CLIP objective instead. However, these approaches are limited to minor edits of input images and cannot generate "from scratch." A recent observation is that a similar gradient-based latent optimization approach applied to image tokenizers with extremely compressed latent representations, such as TiTok (Yu et al., 2024), does succeed in generating high quality images (Lao Beyer et al., 2025). However, to the best of our knowledge, our work is unique in leveraging highly compact ordered and discrete representations to perform efficient latent space exploration via tree search rather than continuous optimization.

**Guidance in diffusion.** Diffusion (Ho et al., 2020) supports test-time conditional sampling without re-training (Ho & Salimans, 2021; Dhariwal & Nichol, 2021; Bansal et al., 2023), and has also been widely used for robotics applications (Chi et al., 2023). Our approach's support for flexible test-time objectives may be reminiscent of loss-guided diffusion (Song et al., 2023). However, guided diffusion using arbitrary objective functions can be challenging to implement, as there is no access to the final "clean" sample during intermediate diffusion steps, yet the desired objective is typically defined only for clean, not intermediate (noisy) samples. In contrast, our search does not suffer from this problem, as the autoencoder's output space matches the input domain of the objective.

## 5 DISCUSSION

Model-based motion planning methods in robotics often optimize expert-designed objective functions, providing a large degree of flexibility and control over robot behavior. However, only powerful priors learned from large scale data can provide the generalization ability and real-world robustness needed for increasingly open-ended tasks such as autonomous driving or manipulation. To unify the strength of both approaches, we view motion planning as search in the latent space of a deep trajectory autoencoder. In particular, we design an environment-conditioned autoencoder with a latent representation that is so compact and structured that a simple greedy search over tokens can be used to successfully plan according to an arbitrary objective function.

We believe that robotics applications are positioned especially well to take advantage of this framework due to the prevalence of useful objectives and constraints. In the task of self-driving, objectives such as waypoint and route following, or constraints such as limits on acceleration and jerk for comfort and safety are common examples. Although we do not explore them in this paper, we envision applications in other areas of robotics such as manipulation, where this framework can be useful in generating behaviors that conform to a certain task specification, reach a desired goal, or incorporate other user preference or domain knowledge specified in the form of an objective function.

## LARGE LANGUAGE MODEL DISCLOSURE

During paper writing, LLMs were used for minor wording suggestions and improvements. LLMs were also used for assistance with writing the plotting code used to generate the visualizations shown in this paper.

LLMs were also used to aid in literature review, and for occasional assistance writing small parts of code used for our experiments.

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

# A    APPENDIX

## A.1    CONDITIONAL AUTOENCODER MODEL AND TRAINING DETAILS

**Trajectory and environment tokenization.** We use a similar strategy to MTR (Shi et al., 2022). Agent trajectories are augmented with additional metadata (object type, timestep index, heading angle, and relative position between consecutive samples) and encoded with a PointNet architecture (Qi et al., 2017). Road edges and road lines are split into 20-point segments and also encoded using a small PointNet-style network. In contrast to the original MTR, we encode each feature the local coordinate frame placed at its initial pose (for agent trajectories) or center of mass (for road geometry).

**Transformer models.** The environment encoder, taking as input the tokenized static world and agent history representations, uses *local self attention* layers to improve efficiency and incorporate a spatial locality bias. We again closely follow MTR (Shi et al., 2022) in this regard.

Our trajectory encoder and decoder make use of blocks of causal self-attention followed by cross-attention. In the case of the encoder, the queries are initialized from the following tokens: (a) the token encoding of the full input trajectory, (b) learnable positional encodings for the latent tokens and (c) one or more additional register tokens (Darcet et al., 2024) (not shown in the diagram in Figure 4). Even though the number of key/value tokens produced by the environment encoder may be large, we do not use any form of local attention or geometric aggregation, since cross-attention wrt. our small number of query tokens is sufficiently fast.

The latent tokens are then processed by the bottleneck consisting of projection to the desired token dimensionality ($D = 3$ in our experiments) and noise injection (during training) or hard quantization (at test time). After projecting back to the transformer feature dimensionality, the decoder processes the latent tokens with the same design as the encoder, noting that we again include one or more register tokens with learnable positional embedding. Finally, we regress the final trajectory from one of the register tokens using a small MLP.

In both our trajectory encoder and decoder's self-attention layers, causal masking is enforced between the latent tokens, while register and input trajectory tokens may attend to all tokens.

The following hyperparameters are shared across the environment encoder model, the trajectory encoder, and the trajectory encoder:

- Layers: 6
- Heads: 8
- Transformer feature dimensionality: 256
- Feed-forward MLP width: 2048

**Token dropout schedule.** During training, we perform token dropout with $50\%$ probability. If dropout is enabled, we choose the number of tokens to keep according to an exponential schedule with probabilities proportional to $(1/2)^{N_{\text{drop}}}$, where $N_{\text{drop}}$ denotes the number of tokens to drop.

**Other training details.** We train our models for around 30 epochs, using the AdamW optimizer (Loshchilov & Hutter, 2019) with a batch size of 64 and learning rate of $10^{-5}$. For adaptive soft quantization, we set $\gamma = 0.9995$ and $\Delta\sigma = 0.01$.

## A.2    DETAILS ON PLANNING OBJECTIVE FUNCTIONS

Our framework supports general objectives of the form

$$f(\mathcal{T}_{\text{pred}}, z) \mapsto \mathbb{R}, \tag{4}$$

where $\mathcal{T}_{\text{pred}} = \text{Dec}(z, \mathcal{E})$ refers to the decoded trajectory, and $z$ is the tokenized representation. Let $n$ denote the length of the token sequence $z$, which starts at $n = 1$ at the beginning of the search, increasing to $n = N$ once the maximum depth is reached. In our experiments, we consider objectives of the following form, which penalize high predictive variance.

$$f_g(\mathcal{T}_{\text{pred}}, z) := g(\mathcal{T}_{\text{pred}}^{(\mu_{xy})}) + \lambda \mathbb{1}[\mathcal{T}_{\text{pred}}^{(\sigma_{xy})} > \sigma_{\max}(n)]. \tag{5}$$

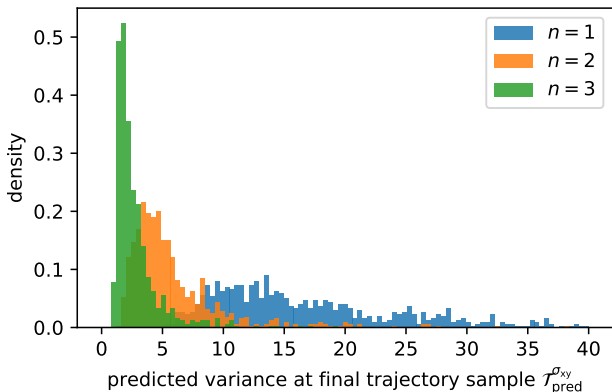

Figure A1: **Predicted variance decreases with increasing number of tokens.** Histogram of predicted variance values illustrates need for token-sequence-length-dependent variance threshold during search.

Here, $\mathcal{T}_{\text{pred}}^{(\mu_{xy})}$ refers to the decoder's predicted mean of the trajectory. We use $\mathcal{T}_{\text{pred}}^{(\sigma_{xy})}$ to denote the magnitude of the predicted covariance for the final sample of the trajectory, which we check against the threshold $\sigma_{\max}(n)$ to impose a heavy penalty $\lambda \gg g(\mathcal{T}_{\text{pred}}^{(\mu_{xy})})$. We find that an uncertainty threshold which depends on the token sequence length is beneficial, as the distribution of the predicted variance depends strongly — as expected — on the length of the latent token encoding (see Figure A1). This leaves us to freely choose the cost function $g$ based on the desired target application.

**Left turn maneuver optimization.** The heading $\theta[i]$ along each segment $i$ of the polyline defined by the current candidate trajectory $\mathcal{T}_{\text{pred}}^{(\mu_{xy})}$ is first computed using finite differences. We can then compute the total cumulative heading change in the counterclockwise direction as $\text{CCW} = \sum_i \max(0, \theta[i+1] - \theta[i])$. This allows us to define the very straightforward cost function

$$g_{\text{left-turn}}(\mathcal{T}_{\text{pred}}^{(\mu_{xy})}) := -\min\{\text{CCW}, \theta_{\min}\}, \tag{6}$$

which encourages turns with a leftward heading change of at least $\theta_{\min}$ (which we set to $\frac{\pi}{4}$). In addition to Table 3, we present visualizations of the left turn maneuver optimization in Figure A2. We highlight that the fact that token search is not always able to achieve the desired heading change is a desirable property: as shown in Figure A2c, our examples include cases where left turns are illegal or impossible.

We also highlight that the ultimate heading change achieved by our token search is *not* simply clustered around the threshold $\theta_{\min}$, indicating that token search find solutions that align with the correct road geometry rather than blindly optimizing the objective which, if used on its own to optimize a less robust trajectory representation, would be far too simple to produce correct behavior.

**Speed reduction.** Our choice of $g$ is again very simple:

$$g_{\text{slowdown}}(\mathcal{T}_{\text{pred}}^{(\mu_{xy})}) := \max_{i \in I} \max\{0, v_i - v_{\max}\}, \tag{7}$$

with $v_i$ denoting the magnitude of the velocity along the $i$th segment of the trajectory, again computed using finite differences, $v_{\max}$ denoting the maximum speed constraint value ($5\,\text{m/s}$ in our experiment) and $I$ denoting the range of timesteps to apply the constraint over (in our experiment we apply it from $t = 6\,\text{s}$ to the end of the predicted trajectory at $t = 9\,\text{s}$).

Again, token search does not enjoy a 100% success rate. However, we argue that this behavior is desirable, allowing even very naively specified objective functions to produce reasonable real-world behavior thanks to the decoder serving as a form of "learned guard rails." Indeed, we show in Figure A3c that our framework makes a best effort to increase deceleration while maintaining smooth and safe longitudinal jerk and acceleration values.

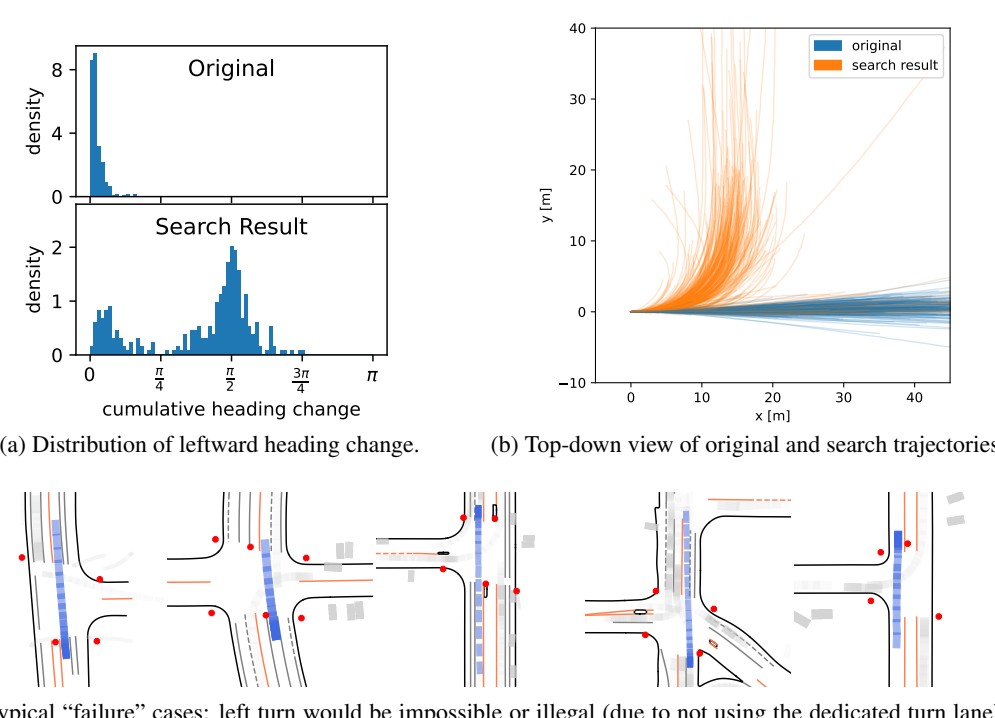

(a) Distribution of leftward heading change.

(b) Top-down view of original and search trajectories.

(c) Typical "failure" cases: left turn would be impossible or illegal (due to not using the dedicated turn lane). We plot the trajectory found by token search with the left turn objective.

Figure A2: **Search for left-turn maneuver.** 339 scenarios filtered by agent's proximity to several stop signs, while moving straight.

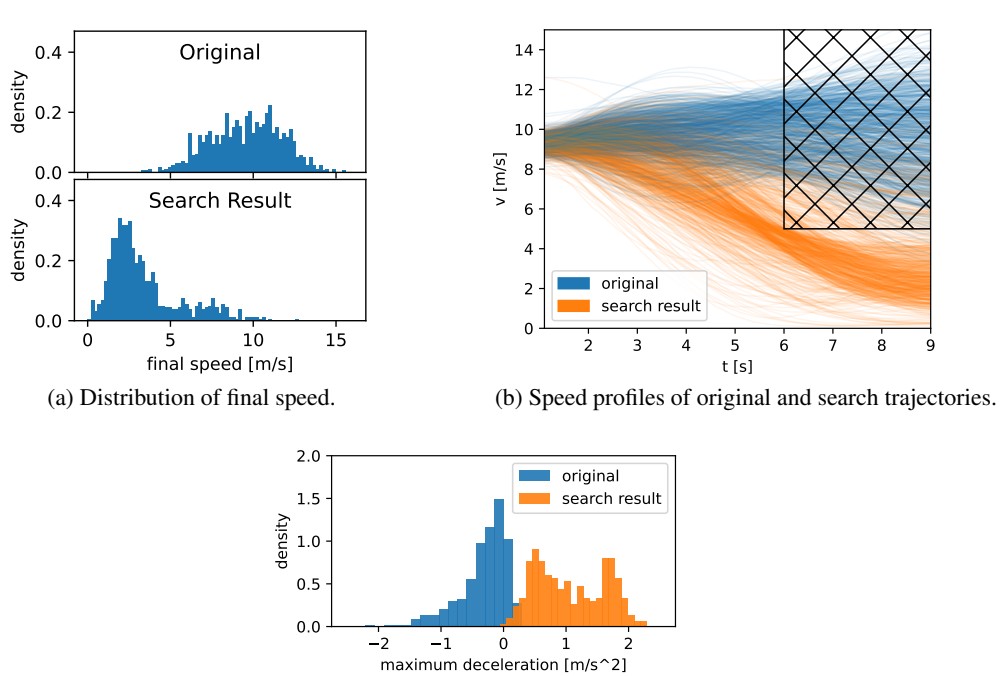

(a) Distribution of final speed.

(b) Speed profiles of original and search trajectories.

(c) Histogram showing maximum *deceleration* of in scenarios where search with speed reduction objective failed to achieve the desired speed reduction. Note that original scenarios are almost exclusively cases with no deceleration or even strong acceleration, and search is able to significantly increase the amount of deceleration.

Figure A3: **Slow-down maneuver optimization.** 794 scenarios filtered by agent's starting speed.

## A.3 MULTI-AGENT MODEL DETAILS

To encode multi-agent trajectories jointly, we first encode each agent $a$'s trajectory $T_a$ individually, using the same architecture as used in our single-agent conditional autoencoder $\text{Enc}_{\text{SA}}$ and $\text{Dec}_{\text{SA}}$:

$$\mathbf{y_a} := \text{Enc}_{\text{SA}}(T_a, \text{center}_a(\mathcal{E})) \quad \text{and} \quad (T_a)_{\text{pred}} := \text{Dec}_{\text{SA}}(\hat{\mathbf{y}}_\mathbf{a}, \text{center}_a(\mathcal{E})). \tag{8}$$

Here, $\text{center}_a(\mathcal{E})$ refers to a geometric transformation of the environment into the agent-centric frame of agent $a$. We now further process the features $\mathbf{y_1}, \ldots, \mathbf{y_a}$ with a second-stage encoder $\text{Enc}_{\text{MA}}$ to produce the latent representation $z$:

$$z = \text{Enc}_{\text{MA}}(\{\mathbf{y_1}, \ldots, \mathbf{y_a}\}, \mathcal{E}) \quad \text{and} \quad \hat{\mathbf{y}}_\mathbf{a} = \text{Dec}_{\text{MA}}(\hat{z}, \mathcal{E}), \tag{9}$$

where $\hat{z}$ refers to the decoder input after applying the bottleneck downprojection, soft or hard quantization, and projection back to the decoder input dimensionality. To preserve invariance of our latent token representation with respect to agent ordering and global coordinate transformations, we implement $\text{Enc}_{\text{MA}}$ and $\text{Dec}_{\text{MA}}$ using a transformer, and incorporate environment information from $\mathcal{E}$ in a way that allows us to preserve the desired invariances. In particular, for each agent $a$, we compute an encoding $\mathbf{p_a}$ of the position of all other agents in agent $a$'s frame using a PointNet-style encoder. The feature $\mathbf{p_a}$ can now act as a positional embedding added to $\mathbf{y_a}$ before feeding it into the transformer. Similarly, we initialize the tokens fed to the transformer model $\text{Dec}_{\text{MA}}$ with $\mathbf{p_a}$, in order to decode each $\hat{\mathbf{y}}_\mathbf{a}$.

## A.4 LLM EXPERIMENT DETAILS

**Precomputed token dataset.** Since the purpose of the language understanding experiment is to detect whether our multi-agent conditional autoencoder learns a semantically rich representation, we do not allow any updates to the trajectory encoder while fine-tuning the language model. Therefore, we start by tokenizing the full WOMD-Reasoning (Li et al., 2025) training set. We additionally include the environment encoder's output in this dataset, which consists of static world features and dynamic object history. Note that we use the ego agent's coordinate frame to encode the environment. We also transform object histories into the ego agent's frame.

**LLM fine-tuning.** We perform supervised fine-tuning (SFT) of the Qwen3-4B-Instruct-2507 (Qwen Team, 2025) model with LoRA (Hu et al., 2022). During fine-tuning we train small adapters to embed our low dimensional tokens $z$ and the conditioning information from $\mathcal{E}$ into the LLM's token space. The tokens $z$ are adapted using a small two-layer MLP. Static world features are taken directly from the (frozen) environment encoder and projected to the LLM's embedding space with a single linear layer. Object histories using an LSTM applied over each object's 11 history observations.

Since we use an instruction-tuned variant of Qwen, we adhere to a chat template of the following form during both training and sampling:

| | |
|---|---|
| Priming prompt | The following information describes a driving scenario: |
| Environment encoding | *[tokens projected from environment encoder]* |
| Agent history information | *[tokens encoded by LSTM]* |
| Latent tokens $z$ | *[tokens encoded by MLP]* |
| Question | `<user>`[Question]`</user>` |
| Answer | `<assistant>`[Answer]`</assistant>` |

**Evaluation.** We follow the protocol of WOMD-Reasoning (Li et al., 2025) and evaluate five different question answering and language translation metrics on a subset of 1000 randomly sampled validation examples: ROUGE-L (Lin, 2004), BLEU, with maximum order 4 and using word-level tokenization (Papineni et al., 2002), METEOR (Banerjee & Lavie, 2005), CIDEr (Vedantam et al., 2015), and SPICE (Anderson et al., 2016).

