# OpenReview forum: "Robotics in Representation Space: Learned Latents Meet Composable Costs"
_ICLR.cc/2026/Conference — Submitted to ICLR 2026_

### Official Review · Reviewer_7r5J · 2025-10-30

**Soundness:** 2
**Presentation:** 1
**Contribution:** 2
**Rating:** 2
**Confidence:** 2

**Summary:**

The authors investigate the applications of using a compressed latent representation in the context of motion planning. Their work is motivated by prior research suggesting that a strong decoder model can reduce the required latent dimensions to reconstruct the target data. They propose a quantized latent representation that they suggest improves when learned using an adaptive noise distribution. Their experiments are on the Waymo Open Motion Dataset, where they show the re-usability of their latent features by analyzing the re-usability of the features in capturing motion meaning and reconstruction quality.

**Strengths:**

> The motivations for the author's research are sound. Classical motion planning algorithms do not scale well to extensive dimensional data, so using low-dimensional latent representations to enable their use makes sense

> The application of a compressed latent space is an interesting idea, and the authors' research has practical applications.

> Figure 4 illustrates the model architecture proposed by the authors.

**Weaknesses:**

We found it challenging to follow the submitted paper draft. Many sections require rewriting for clarity, where even small details are not serving to communicate the author's point. The title suggests a generalist representation approach to robotics, but the authors only apply their autoencoder approach to a self-driving data set. Much of the paper discusses robotics in a general sense, but since the experiments focus only on autonomous driving, I consider this misleading to readers. It would serve the paper better to focus on this application rather than use general-sounding language (e.g., "robotics tasks" should really be "self-driving").  The author's method may have applications in other domains, but as they do not demonstrate this in the experiments, it might be better to discuss this only in the conclusion as future work.

Furthermore, sections 2 & 3 in particular would benefit from extensive rewriting. It could be helpful to clarify how the latent codes are combined with classical motion planning in Section 2. Section 3 should be clearer that these are the experiments. Even just changing the section header to "Experiments" or clearly stating the experiments' objectives in the preamble would already be an improvement.

The writing also makes it difficult to determine the paper's significant contributions or its advantages. For example, Table 2 shows several models surpass the author's model performance (e.g., DriveGPT has 0.5240 minADE_6 vs the author's 0.6415). It is mentioned that the author's latent space is notably smaller; what about the decoder model's capacity in comparison? Do the baseline models use larger or smaller decoders?. Likewise, quantization is emphasized as necessary, but the reported metrics in Tables 1 & 5 suggest that excluding quantization leads to better performance.  Similarly, it is not clear from the discussion in Table 4 why matching Motion-LLaVA is essential, and should be clearly stated in the writing.  Given these issues, we believe the paper as a whole needs improvements not just in writing but also in its empirical evaluations to strengthen its claims.

Comments on Method:
Section 2.1 (Line 14): the quantization scheme described should cite prior research that reports a closely related (near-identical) approach [1]. This work was not mentioned in the related work.

[1] Mentzer, Fabian, et al. "Finite Scalar Quantization: VQ-VAE Made Simple." The Twelfth International Conference on Learning Representations.

> Figure 2: This idea of nosing latent embeddings has been considered in other research. VAEs do this with the reparameterization trick, yet this feature is prominently established throughout the author's work (e.g. Figure 1 shows how vital this regularization trick is). If this adaptive noising approach is beneficial, it would be better justified if alternative noising schemes (i.e. a stochastic latent variable such as from a VAE) were compared against.




Writing Opinions

> The introduction as a whole could be improved for clarity. We suggest restructuring the focus to the limitations of motion planning and the benefits of autoencoders to address these problems, rather than discussing autoencoders and treating motion planning as secondary.

> Line 036:  "autoregressive" is ambiguous in this sentence; make it more specific.

> Line 038: "autoencoder" is not set up well in this paragraph and is confusing to read about as written.

> Line  488 - 492:  Combine content into a single paragraph

**Questions:**

Q1: Have the authors considered training a decoder that only relies on the latent variable for reconstruction? Is the reason the latent space can be compressed so much because of these additional inputs?

Q2: What is the red square input to the decoder model?

Q3: Why did the author use the WOMD dataset? In what other domains could this method be employed?

Q4: We find the results in Figure interesting. What do the authors believe explains this observed re-usable aspect of their latent representation?

---

> ### Author Response · Authors · 2025-11-24
> **Response to Reviewer 7r5J (1/2)**
>
> Thank you for your review. We find your questions and concerns regarding presentation helpful, and agree that the framing of our work in the introduction could potentially be stronger by focusing on the motion planning problem from the start. To address any confusion that this may have caused, we would like to provide an alternative framing of our work below:
>
> A very popular classical approach to motion planning consists of concatenating trajectories selected from within a finite, predefined library of maneuvers [0, 1]. This library is typically static and consists of feasible trajectories that do not take into account external observations (such as obstacles). Therefore, if obstacle avoidance or other environment-dependent behavior is needed, it must be implemented explicitly as part of the maneuver selection algorithm. In contrast, our proposed conditional autoencoder can be seen as a mechanism for learning a *context-dependent* maneuver library: since the decoder is conditioned on the environment, and trained to reconstruct trajectories from a very low-dimensional representation, it effectively learns the space of valid trajectories in the particular given environment, automatically excluding solutions inconsistent with road geometry or other agents' observed behavior.
>
> Planning with this maneuver library is conceptually identical to the case of static libraries: we may enumerate every candidate trajectory and select the best according to our desired objective and constraints. However, since in our case the library is dynamic and aware of the context, the search space is automatically reduced to trajectories which are consistent with the environment (i.e. behavior consistent with the road geometry and other agents in our Waymo examples), greatly simplifying the design of downstream cost functions.
>
> Furthermore, our dynamic maneuver library has a hierarchical structure due to the causal ordering constraint and nested dropout applied to the latent representation. This enables very simple and efficient greedy search to succeed in finding the desired trajectories from the library even when it is of large size (from hundreds of potential candidates in our single-agent examples to hundreds of thousands in the multi-agent examples). In summary, under this alternative framing of our work, our paper provides a method to learn hierarchical, context-dependent maneuver libraries for motion planning.
>
> ---
>
> To address your questions and comments, we have provided answers below.
>
> * **Q1: "Unconditional" autoencoder**
>
> Indeed the high degree of compression in our case is enabled by the additional conditioning inputs. We do not train an unconditional autoencoder, since our goal is to learn a *context-dependent* maneuver library rather than a static one. In particular, the conditioning is critical in the proposed planning framework as it ensures that the decoded latents map to behavior which is valid in the given context – i.e. we get collision avoidance, lane following, etc. "for free" from the decoder without having to explicitly include it in our cost function.
>
> * **Q2: Model architecture clarification**
>
> The red square input to the decoder model represents a learnable latent query token that is updated by the attention layers to aggregate information used to produce the final trajectory reconstruction. It is equivalent to a CLS or register token commonly used in other transformer architectures.
>
> * **Q3: Choice of dataset**
>
> We use the Waymo Open Motion Dataset (WOMD) as it contains high quality, long-horizon examples including rich behavior (e.g. object interactions, complex road geometry). We find that this provides a good testbed to showcase the proposed decoupling between "high-level" behavior synthesis according to user-defined objectives (as enabled by the proposed latent space search) and "low-level" adherence to rules of the road (as learned by the decoder, via the task of reconstructing trajectories from a highly compressed representation).
>
> Another domain where we would like to apply our method is manipulation. For example, one could train an image or video-conditioned autoencoder of manipulator trajectories, and then perform tasks according to arbitrary test-time objectives via latent search, following the same framework we present in our paper.
>
> * **Q4: Figure 5 results on behavior transfer via token complexity**
>
> Please let us know if you were referring to a figure other than Fig. 5. In the case of Fig. 5, we believe that it demonstrates the value of encoding trajectories conditionally on the environment (see also our answer to your Q1). In particular, our intuition is that the highly compressed bottleneck in the autoencoder forces the model to learn stable/transferable representations because they encode the trajectory efficiently.
>
> ---
> **[continued below]**

---

> > ### Author Response · Authors · 2025-11-24
> > **Response to Reviewer 7r5J (2/2)**
> >
> > * **Similarity to FSQ**
> >
> > We do use a quantization scheme very similar to that used in FSQ. However, while FSQ applies quantization at training time, we apply it purely at test-time (replacing it with adaptive noise injection at training time). In our opinion, this is substantially different and we find it to be crucial to enable convergence when using the proposed low-dimensional representations (compared to quantizing at training time and estimating gradients straight-through). In any case, we agree that FSQ should be cited, and its relation to our quantization scheme be clarified in the paper.
> >
> > * **Applicability of VAE**
> >
> > Since the regularization scheme used in VAEs encourages normally distributed, continuous latents, the VAEs latent space does not lend itself to the same exploration via discrete graph search as proposed by our method. Therefore, we believe that a meaningful comparison of motion planning using a VAE-based conditional autoencoder would be difficult.
> >
> > Please do not hesitate to let us know there are any unresolved or further questions!
> >
> > ---
> >
> > References:
> > * [0] Frazzoli, Emilio, Munther A. Dahleh, and Eric Feron. "Maneuver-based motion planning for nonlinear systems with symmetries." IEEE Transactions on Robotics 21.6 (2005): 1077-1091.
> > * [1] Pivtoraiko, Mihail, Ross A. Knepper, and Alonzo Kelly. "Differentially constrained mobile robot motion planning in state lattices." Journal of Field Robotics 26.3 (2009): 308-333.

---

### Official Review · Reviewer_UqUz · 2025-11-01

**Soundness:** 2
**Presentation:** 3
**Contribution:** 2
**Rating:** 4
**Confidence:** 3

**Summary:**

This paper proposes a method to leverage recent deep learning advances in motion planning for robotics with traditional search and optimisation based planning algorithms. The key contribution is the ability that the method provides to optimise arbitrary objectives specified by the user without having to retrain or fine-tune these networks, simply by relying on the pretrained autoencoder. The results of this method are demonstrated on the Waymo Open Motion dataset for both motion prediction and behaviour generation.

**Strengths:**

- The approach aims at utilising one of the biggest insights from the past few years regarding the impact of large-scale real-world data to learn useful priors about the world, and channeling it to robotics to enable the use of traditional search algorithms in robotics.
- Claims on composability of objectives in the driving scenario and greedy search versus learned encoder are well substantiated.
- The main advantage of this work is the cheap adaptation tailored to the user’s objective, without having to perform additional training.

**Weaknesses:**

- One of the main weaknesses of the paper is the reliance of the empirical study on a single dataset, the Waymo Open Motion dataset. This makes it difficult to assess the generalisation of this approach to other robotics domains.
- A key advantage of this approach described in the paper is the ability to use these compact latent tokens in conjunction with search algorithms. However, the only search studied in this paper is greedy search. Again, the issue of generalisation comes up in how this method would work with other search strategies like beam search or Monte-Carlo Tree Search.
- When talking about real-world domains, a critical aspect is that of reliability and safety, which is not adequately addressed in this paper. Given that the paper attempts to provide a cheap way to achieve different user-guided behaviours, guarantees of reliability and safety, or the lack thereof, should be addressed.

**Questions:**

This work positions itself as one that gives a best-of-both-worlds scenario for foundation models for robotics and self-driving, and search and planning algorithms. However, since it is heavily dependent on having a high quality autoencoder, which in turn depends on having a large diverse dataset, could the authors shed some light on how such a method might be used in a low-data regime? No new results are required, just an intuitive understanding on how one might leverage the proposed method in robotics domains where large datasets are not available.

---

> ### Author Response · Authors · 2025-11-24
> **Response to Reviewer UqUz**
>
> Thank you for the valuable concerns and questions raised in your review. We have attempted to address your points below.
>
> * **Datasets**
>
> We chose the Waymo Open Motion dataset (WOMD) for our experiments, as we believe that the self-driving domain can provide a valuable testbed for our approach, and WOMD provides high-quality and diverse data. In particular, the driving scenarios contained in WOMD contain rich and complex interaction with the environment, providing an ideal opportunity to showcase the proposed decoupling between "high-level" behavior synthesis according to user-defined objectives and "low-level" adherence to rules of the road.
>
> We would also like to highlight our experiments on joint trajectory encoding of multi-agent scenes. While these experiments still make use of WOMD, they demonstrate scaling of our approach to a substantially more complex domain. Furthermore, we demonstrate generation or editing of scenarios according to user-defined objectives while maintaining jointly consistent behavior as a promising and novel application beyond behavior prediction or planning.
>
> * **Search methods**
>
> Our focus on greedy search is a deliberate choice to demonstrate the effectiveness of our causally-ordered latent representations. Indeed, since greedy search is one of the most naive choices of search algorithm, we argue that its success in synthesizing the desired behavior for a variety of objectives is a strong indication of the success of causally-ordered representations in enabling tractable latent space exploration. Nevertheless, we agree with the reviewer that it would be beneficial to include results with stronger search methods. We have therefore implemented beam search and report results for reconstruction below. We observe that for small increases in beam size, we obtain relatively significant improvements in accuracy. At the same time, the structure of our latent space leads to rapidly diminishing returns when further increasing the beam width, as expected.
>
> | Beam size | ADE (reconstruction) |
> |:---------:|:-----:|
> | 1         | 0.386 |
> | 2         | 0.333 |
> | 4         | 0.310 |
> | 8         | 0.304 |
> | 16        | 0.298 |
>
> We also repeat the maneuver optimization experiments from Table 3 with beam search and report the updates success rates. These results reveal that objectives other than reconstruction can also benefit from modest increases in beam size.
>
> | Search algorithm | Left turn (road edge contact) | Speed reduction (road edge contact) |
> |:----------------:|:-----------------------------:|:-----------------------------------:|
> | Ground truth     | 0% (0%)                       | 0% (0.76%)                          |
> | Greedy           | 75.5% (0%)                    | 63.2% (0.13%)                       |
> | Beam (size 2)    | 76.7% (0%)                    | 73.3% (1.39%)                       |
> | Beam (size 4)    | 79.9% (0%)                    | 73.5% (1.76%)                       |
>
> * **Safety**
>
> While enhanced safety characteristics compared to other learning-based motion planning approaches such as imitation learning is not a focus of our work, we agree that it is important to mention. Viewing the search in our decoder's latent space as trajectory selection (from within a dynamic, context-dependent behavior library), it is straightforward to filter any trajectories that can be deemed infeasible or unsafe with an explicit model-based check (the check can just be incorporated as part of the search objective). While this framework provides additional control compared to typical learning-based methods, our approach cannot guarantee the existence of feasible and safe trajectories for arbitrary inputs beyond empirical results.
>
> * **Low-data regime**
>
> For applications in domains where a suitable dataset is unavailable, we believe that online data collection via reinforcement learning could provide a potential solution for domains where a simulator is instead available. In particular, future work may consider using reinforcement learning to learn a policy conditioned on a learned, causally ordered, quantized latent representation (similar to our conditional autoencoder) which can later be optionally guided by "plug-and-play" user-defined objectives via latent search and without requiring any further training (same as our proposed method).

---

### Official Review · Reviewer_2tWV · 2025-11-07

**Soundness:** 3
**Presentation:** 3
**Contribution:** 3
**Rating:** 4
**Confidence:** 4

**Summary:**

The paper proposes planning as search in a learned discrete latent space. A conditional transformer autoencoder is trained on WOMD to produce causally-ordered, low-dimensional, quantized tokens for trajectories; at test time, the decoder serves as a strong generative prior while a greedy tree search over tokens optimizes arbitrary user objectives (e.g., “turn left”, “slow down”). The model also explores multi-agent tokenization and shows that tokens convey semantic behaviors (token swapping, “library of behaviors”). Key ingredients include adaptive soft quantization at the bottleneck, nested dropout to enforce coarse-to-fine token order, and hard quantization during search. Experiments cover reconstruction, prediction via variance-minimizing search, guided maneuver generation, and a multi-agent VQA proxy by feeding tokens to an LLM.

**Strengths:**

1. Crisp conceptual unification. The paper cleanly separates objective choice at test time from trajectory realism via the decoder, and leverages an ultra-compact latent space to make tree search feasible.

2. Simple, well-motivated mechanics. Adaptive noise (“soft quantization”) and nested dropout jointly encourage discrete, ordered latents; this is elegant and empirically helpful (Fig. 2).

3. Compelling token semantics. Token swapping and “behavior libraries” show transferable high-level behaviors, not just compression (Fig. 5).

4. Test-time flexibility. Greedy token search handles different objectives without retraining (left-turn, slow-down), while the decoder enforces map-consistency (edge-contact ≈0%).

5. Multi-agent extension. Joint tokenization (up to 8 agents) plus simple goal objectives yields qualitatively consistent interactions; tokens also aid an LLM on WOMD-Reasoning.

**Weaknesses:**

1. Planning evidence is under-baselined. Section 3.4 reports maneuver “success rate” and edge-contact against the unmodified scenario, but no planning baselines (trajectory optimization, sampling-based planners, MPPI/IT-MPC, or diffusion planners) are included. As a result, it’s hard to judge optimality/efficiency and where search in latents sits among standard planners.

2. Scalability beyond toy latent sizes is unclear. All strong results use very small spaces (e.g., N=3 tokens, D=3 dims, Nlevels=2), where greedy cost is merely N·Nlevels^D = 3·2^3 = 24 decoder calls. The paper acknowledges this efficiency (≈115 traj/s on RTX 6000 Ada), but does not test larger N/D/Nlevels where branching explodes and greedy likely degrades. A scaling study is essential.

3. Objective realism and multi-objective tradeoffs. The showcased costs (cumulative left heading; target final speed) are simple and largely single-objective. Real planners optimize safety/comfort/progress and constraints (jerk, acceleration, distance to agents). The method’s greedy, token-by-token decisions might be myopic under competing objectives; no experiments probe this.

4. Reliance on decoder variance as a guard-rail is risky. Prediction and planning searches penalize decoder-predicted variance, but heteroscedastic NNs can be miscalibrated (the paper cites this literature). There are no calibration diagnostics (PIT, ECE, risk-coverage), so search could be steered by poorly calibrated uncertainty.

5. Limited safety metrics for interactions. Planning reports edge-contact with static map; dynamic-agent collisions, min-distance, TTC are not measured. Multi-agent results are mainly qualitative (Fig. 6) and a language metric table (Table 4) that doesn’t evaluate physical safety of generated interactions.

6. Causal ordering assumption not validated against optimization. The paper assumes nested-dropout + causal masking yields a coarse-to-fine order that aligns with greedy objective optimization. Evidence is indirect (reconstruction improves with more tokens; Fig. 3). There’s no analysis showing early tokens control “coarse” semantics relevant to downstream costs.

7. Open-loop feasibility and dynamics. The decoder produces kinematically smooth trajectories, but there’s no explicit vehicle dynamics or closed-loop execution. It’s unclear if generated plans remain feasible/stable under tracking noise or model errors.

**Questions:**

1. Scalability: What is planning quality/runtime as N, D, Nlevels increase? Have you tried beam search or MCTS in latent space, and how do they compare to greedy for complex objectives? (Complexity ≈ N·Nlevels^D per greedy layer.)

2. Baselines: Can you add at least one trajectory optimizer (with the same costs) and one sampling/diffusion planner on the same WOMD slices used in Table 3?

3. Safety metrics: For planning and multi-agent generation, can you report collision rate, min distance, jerk/accel, and rule/route compliance in addition to edge-contact and success rate?

4. Uncertainty calibration: How calibrated is the decoder’s variance? Any risk-coverage or PIT plots before using it to gate search?

5. Causal ordering verification: Can you intervene on individual tokens to quantify their semantic scope (e.g., MI between token i and global maneuvers vs. local kinematics), supporting the coarse-to-fine claim?

6. Closed-loop: Have you tested tracking a decoded plan with a simple controller (bicycle or PID) to measure closed-loop feasibility and regret?

---

> ### Author Response · Authors · 2025-11-24
> **Response to Reviewer 2tWV (1/3)**
>
> We would like to thank you for your thorough evaluation and your many valuable suggestions and questions. In the following, we have attempted to address as many of your questions and concerns as we could.
>
> * **Scale of latent space**
>
> **Justification of tiny latent space size.**
> The size of the latent space is indeed very small. However, in our view, this does not mean that our approach is limited to toy problems. We justify this using the results from Table 1. In particular, we find that performing greedy search with just three 3-bit tokens (i.e. N=3, D=3, Nlevels=2) can reconstruct a given trajectory very accurately (outperforming all prediction models by a large margin -- cf. Table 2). This motivates us to conclude that, conditionally on the scenario context (road geometry and agent motion history) our tiny latent space is more than sufficient to represent a diverse enough range of maneuvers.
>
> While scaling to larger latent spaces would potentially allow us to further improve reconstruction accuracy, we believe that an excessively large bottleneck in our conditional autoencoder would lead to an underutilized latent space in which many encodings do not map to valid behavior when decoded. Therefore, planning via latent search would be likely to produce invalid behavior. For the same reason, we would also not expect "token swapping" like shown in Figure 5 to enable meaningful transfer of behavior across scenarios when using larger latent spaces.
>
> **Scaling to larger latent spaces.**
> As you point out, scaling to larger latent spaces can additionally lead to increased computational cost of search. However, we note that when scaling the number of tokens, the computational complexity of greedy or beam search would scale only linearly, while the overall size of the latent space would scale exponentially. To demonstrate the effectiveness of scaling the latent size by increasing the number of tokens while keeping each token's dimensionality fixed (and thus increasing search complexity only linearly), we turn to the multi-agent problem from Section 3.5, since joint trajectory encoding can benefit from a larger representation. The following table shows the results of reconstruction-guided latent search with a new multi-agent trajectory autoencoder trained with up to $n=6$ tokens while keeping the token dimensionality of three, and applying binary quantization ($N_\mathrm{levels} = 2$).
>
> | num. tokens | ADE (greedy search) |
> |:-----------:|:-------------------:|
> | n=4         | 0.8295              |
> | n=5         | 0.7582              |
> | n=6         | 0.7029              |
>
> **Different search strategies.**
> We have implemented beam search and find that it improves performance when using a small beam size of 2-8. Scaling beyond this beam size leads to only very marginal improvements, as expected due to the ordered structure of our representation. Please refer to our response to Reviewer UqUz, who also asked about alternate search strategies, for more details.
>
> * **Baselines**
>
> **Trajectory optimizer with the same costs.**
> For the results from Table 3, we would like to highlight that the cost functions used are extremely simple -- for example, for left turn maneuver optimization, we maximize the cumulative leftward heading change until reaching some minimum desired cumulative heading change (see appendix section A.2). There is no map-specific route or waypoint at all. As such, the main purpose of our experiments in Table 3 is to show that even such overly simple costs can produce meaningful behavior. If we attempted to use the same cost function in a trajectory optimizer, it would fail to produce meaningful results (e.g. immediately turning without regard for road geometry or other agents) unless we add many other additional costs/constraints to account for kinematic/comfort limits, road geometry and rules of the road, non-ego agent modeling and collision avoidance, etc.
>
> We face a similar challenge in implementing a sampling-based planner -- it would require us to essentially implement a full model-based self-driving stack in order to generate the appropriate constraints. None of this is required in our latent search based approach, since the decoder has effectively learned a good approximation for the space of valid driving behavior.
>
> * **Safety metrics**
>
> Since evaluation in the current version of the paper is open-loop, we find that collision rate cannot be fairly evaluated without allowing the agent to respond to new observations. We agree that these are very important metrics, so **we have redone the intersection maneuver experiments fully in closed-loop and with continuous replanning, for which we report collision and smoothness metrics at the bottom of our response (third comment).**
>
> ---
>
> **[continued below]**

---

> > ### Author Response · Authors · 2025-11-24
> > **Response to Reviewer 2tWV (2/3)**
> >
> > * **Uncertainty calibration**
> >
> > Following your suggestion, we have produced a risk-coverage analysis using ~1500 randomly selected validation scenarios. The results indicate that the model's variance output is indeed predictive of the reconstruction error. This holds when not using any test-time quantization as well as when aggressively quantizing to $N_\mathrm{levels} = 2$, as shown below (where FDE refers to the mean absolute deviation of the final sample of the trajectory):
> >
> > | Coverage | FDE (no quant.) | FDE (N_levels = 2) |
> > |:--------:|:---------------:|:-----------------------------:|
> > | 5%  | 0.17 | 0.25 |
> > | 15% | 0.35 | 0.55 |
> > | 20% | 0.50 | 0.75 |
> > | 35% | 0.63 | 0.94 |
> > | 65% | 0.74 | 1.09 |
> > | 80% | 0.83 | 1.23 |
> > | 95% | 0.95 | 1.39 |
> >
> > * **Causal ordering verification**
> >
> > Evidence that tokens control semantics relevant to downstream costs is indeed critical to show. We had intended the results from Table 3, showing increasing success rate with increasing search depth for different objectives, to provide this evidence.
> >
> > We have also performed a more "direct" experiment in line with your suggestion: for a given scenario, we apply perturbations to the $n$th token and compute their effect on (a) the predicted trajectory and (b) the cost value for both the left-turn and the speed-reduction cost. In particular, we compute the average empirical variance experienced under perturbation of the $n$th token of the terminal position and the two cost functions from Section 3.4:
> >
> > |                      | n=1 | n=2 | n=3 |
> > |----------------------|:-----:|:-----:|:-----:|
> > | terminal position    | 217.2 | 120.0 |  54.4 |
> > | left turn cost       | 0.739 | 0.414 | 0.305 |
> > | speed reduction cost | 5.351 | 3.176 | 1.988 |
> >
> > ---
> >
> > **[continued below]**

---

> > > ### Author Response · Authors · 2025-11-24
> > > **Response to Reviewer 2tWV (3/3)**
> > >
> > > * **Closed loop experiments**
> > >
> > > We have run new closed-loop experiments to showcase (a) feasibility of generated trajectories when followed by a bicycle model (b) stability of the model when continuously replanning and (c) a more realistic route-following cost.
> > >
> > > In particular, we use the lane graph provided in WOMD to find intersections and extract both straight and left-turning routes alongside corresponding ground-truth maneuvers. Given such a route, we define a simple route following objective that maximizes route progress (arclength traversed along the route, determined by projecting the terminal position of the plan onto the route) while penalizing excessive terminal distance to the route. We then run the proposed latent search algorithm to synthesize a trajectory from which kinematically feasible steering angle and acceleration are computed (with curvature capped at 0.1). We continuously replan at a fixed rate of 2 Hz. Other agents follow their logged paths for lateral control, while longitudinal control is either replayed as well or performed according to the intelligent driver model (IDM) [0] if needed to avoid rear-ending the simulated agent. We use the Waymax simulator [1] for these experiments.
> > >
> > > Safety and success metrics. Success is defined as remaining on-route and making forward progress along the route. The logged ground-truth trajectory executes the same maneuver and therefore provides a baseline.
> > >
> > > | Straight scenarios         | Sim result | Log trajectory (reference)|
> > > |:--------------------------:|:----------:|:-------------------:|
> > > | At-fault collision         | 3.6%       | 0%                  |
> > > | Success within 8 seconds   | 70.3%      | 91.7%               |
> > > | Success within 12 seconds* | 94.9%      | N/A                 |
> > >
> > > | Left turn scenarios        | Sim result | Log trajectory (reference) |
> > > |:--------------------------:|:----------:|:-------------------:|
> > > | At-fault collision         | 7.8%       | 0%                  |
> > > | Success within 8 seconds   | 89.7%      | 90.0%               |
> > > | Success within 12 seconds* | 92.3%      | N/A                 |
> > >
> > > (* Object tracks are not available past 8 seconds, so simulation after 8 seconds proceeds without dynamic agents.)
> > >
> > > Comfort metrics: aggregate statistics of maximum acceleration and jerk across all scenarios.
> > >
> > > | Straight scenarios | Simulation result            | Log trajectory          |
> > > |:------------------:|:----------------------------:|:----------------------------:|
> > > |                    | 95th percentile / mean / std | 95th percentile / mean / std |
> > > | maximum lon. accel | 1.60 / 0.75 / 0.57 |  3.63 / 1.65 / 1.26 |
> > > | maximum lon. jerk  | 4.11 / 2.56 / 1.00 | 13.16 / 6.21 / 4.44 |
> > > | maximum lat. accel | 0.27 / 0.13 / 0.09 |  0.61 / 0.31 / 0.19 |
> > > | maximum lat. jerk  | 1.59 / 0.68 / 0.49 |  2.88 / 1.47 / 0.79 |
> > >
> > > | Left turn scenarios | Simulation result            | Log trajectory          |
> > > |:-------------------:|:----------------------------:|:----------------------------:|
> > > |                     | 95th percentile / mean / std | 95th percentile / mean / std |
> > > | maximum lon. accel  | 1.66 / 0.73 / 0.71 | 3.10 / 1.65 / 1.75 |
> > > | maximum lon. jerk   | 2.94 / 1.92 / 2.42 | 14.22 / 7.27 / 11.06 |
> > > | maximum lat. accel  | 2.00 / 1.44 / 0.48 | 4.03 / 2.72 / 0.68 |
> > > | maximum lat. jerk   | 4.08 / 2.27 / 1.08 | 7.01 / 4.38 / 2.16 |
> > >
> > > We have uploaded example simulation videos to the supplementary materials, and are working to incorporate these new results into the main text of the paper.
> > >
> > > ---
> > >
> > > References:
> > > * [0] Treiber, Martin, Ansgar Hennecke, and Dirk Helbing. "Congested traffic states in empirical observations and microscopic simulations." Physical review E 62.2 (2000): 1805.
> > > * [1] C. Gulino, et al. Waymax: An Accelerated, Data-Driven Simulator for Large-Scale Autonomous Driving Research. NeurIPS 2023.

---

### Author Response · Authors · 2025-12-04
**Summary of Discussion**

We are glad that reviewers recognize the value of leveraging structured exploration in latent space to plan with arbitrary objectives: our approach provides a way to "cleanly separate objective choice at test time from trajectory realism via the decoder" (Reviewer 2tWV) enabling "cheap adaptation tailored to the user's objective without having to perform additional training" (Reviewer UqUz).

A common concern was related to limitations in the evaluation of our proposed latent-search planner. We have addressed this concern as follows.
* We have run a series of **new closed-loop experiments** using a simple, yet realistic, route-progress cost.
* Our new closed-loop evaluation ensures kinematic constraints are enforced and enables us to report meaningful safety and comfort metrics. This includes collision rates with other agents and lateral and longitudinal acceleration and jerk. Videos showing the qualitative behavior of our planner in closed-loop have been uploaded as supplementary material, and a [quantitative evaluation against ground-truth trajectories is provided in our response to Reviewer 2tWV](https://openreview.net/forum?id=VFaYukYt6K&noteId=685kRVJtdD). Overall, we observe that **our planner can be successfully deployed in closed loop, complying with our route-following objective while respecting rules of the road and avoiding collisions and executing maneuvers smoothly.**
* We have further demonstrated the effectiveness of our ordered latent representation by considering beam search in addition to simple greedy search. We find that increasing the beam size does lead to improved search results, although our ordered latent representation ensures that gains from larger beam sizes are marginal (see our [response to Reviewer UqUz](https://openreview.net/forum?id=VFaYukYt6K&noteId=qTDI3TDKmP)). This experiment provides additional evidence that the ordered latent representation is key in enabling efficient latent exploration and justifies our choice of greedy search.

We believe that we have carefully addressed all the remaining concerns raised by each reviewer in the respective rebuttal comments. In particular, we would like to emphasize the following.
* Thanks to the environment conditioning, our approach can be used to achieve very high compression of single-agent trajectories, resulting in very small latent spaces while retaining high reconstruction quality. The small latent size does not mean that our approach is not scalable: we show that increasing the number of tokens, which results in an exponential increase to the size of the latent space, can allow our approach to handle more complex domains such as multi-agent trajectories. Yet, the additional computational cost incurred by the latent space search increases only linearly due to the ordered structure of the latent space.
* Our approach may be viewed as a way to learn hierarchically structured, *context-dependent* libraries of maneuvers. As opposed to popular classical motion planning approaches relying on fixed maneuver libraries, our decoder ensures that the library of available behaviors adapts in order to be consistent with the environment (road geometry, other agents), while the proposed greedy search mechanism enables highly efficient exploration of potentially very large such maneuver libraries. We believe this provides a natural and elegant way to leverage learning from large-scale data while enabling highly flexible generation according to user-specified objectives.

We again thank the reviewers for their thoughtful questions which have helped improve our paper!

---

### Meta-Review · Area_Chair_eYV2 · 2026-01-05

**Summary:**

This paper proposes a planning framework that performs search in a learned discrete latent space for autonomous driving. A conditional transformer autoencoder is trained on the Waymo Open Motion Dataset to produce causally ordered, quantized tokens representing trajectories. At test time, greedy or beam search over these tokens enables optimization of user-defined objectives without retraining, while the decoder enforces realism and map consistency. The method demonstrates flexible maneuver generation, prediction, and extensions to multi-agent settings, including qualitative interaction consistency and LLM-based reasoning.

**Reviewer Concerns:**

Reviewer 2tWV questioned whether the planning results are meaningfully evaluated without comparison to standard planning baselines, raised concerns about scalability since all strong results rely on extremely small latent spaces, and asked whether greedy token-wise optimization remains effective under multi-objective or competing costs. Reviewer UqUz expressed concern about generalization, noting that all experiments are conducted on a single dataset (WOMD), and questioned whether the method’s reliance on greedy search limits its applicability compared to other search strategies such as beam search or MCTS.

**Reviewer Scores:**

For Reviewer 2tWV, the concern “Planning evidence is under-baselined… no planning baselines (trajectory optimization, sampling-based planners, MPPI/IT-MPC, or diffusion planners) are included” was not fully resolved.  This concern remains insufficiently answered and would likely limit any score increase.

For Reviewer UqUz, the question “the only search studied in this paper is greedy search… how this method would work with other search strategies like beam search or Monte-Carlo Tree Search” was partially addressed. This leaves the generalization of the planning framework across search strategies only partially resolved, and the reviewer would likely maintain their original rating rather than increase it.

---

### Decision · Program_Chairs · 2026-01-26

Reject